# Uranium Carbide Fibers with Nano-Grains as Starting Materials for ISOL Targets

**DOI:** 10.3390/nano10122458

**Published:** 2020-12-09

**Authors:** Sanjib Chowdhury, Leonor Maria, Adelaide Cruz, Dario Manara, Olivier Dieste-Blanco, Thierry Stora, António Pereira Gonçalves

**Affiliations:** 1C^2^TN, Instituto Superior Técnico, Universidade de Lisboa, Campus Tecnológico e Nuclear, Estrada Nacional 10, 2695-066 Bobadela LRS, Portugal; sanjibbua@gmail.com (S.C.); leonorm@ctn.tecnico.ulisboa.pt (L.M.); adelaide@ctn.tecnico.ulisboa.pt (A.C.); 2CQE, Instituto Superior Técnico, Universidade de Lisboa, Campus Tecnológico e Nuclear, Estrada Nacional 10, 2695-066 Bobadela LRS, Portugal; 3Joint Research Centre, European Commission, P.O. Box 2340, D-76125 Karlsruhe, Germany; Dario.MANARA@ec.europa.eu (D.M.); Oliver.dieste@gmail.com (O.D.-B.); 4CERN—European Organization for Nuclear Research, 23 Genève, CH-1211 Genève, Switzerland; thierry.stora@cern.ch

**Keywords:** nuclear targets, uranium carbides, nano-scale, electrospinning, fibers

## Abstract

This paper presents an experimental study about the preparation, by electrospinning, of uranium carbide fibers with nanometric grain size. Viscous solutions of cellulose acetate and uranyl salts (acetate, acetylacetonate, and formate) on acetic acid and 2,4-pentanedione, adjusted to three different polymer concentrations, 10, 12.5, and 15 weight %, were used for electrospinning. Good quality precursor fibers were obtained from solutions with a 15% cellulose acetate concentration, the best ones being produced from the uranyl acetate solution. As-spun precursor fibers were then decomposed by slow heating until 823 K under argon, resulting in a mixture of nano-grained UO_2_ and C fibers. A last carboreduction was then carried out under vacuum at 2073 K for 2 h. The final material displayed UC_2−y_ as the major phase, with grain sizes in the 4 nm–10 nm range. UO_2+x_ was still present in moderate concentrations (~30 vol.%). This is due to uncomplete carboreduction that can be explained by the fiber morphology, limiting the effective contact between C and UO_2_ grains.

## 1. Introduction

The Isotope Separation On-Line (ISOL) method is a technique used to produce radioactive-ion beams (RIBs) by irradiating a target with a high-intensity/high-energy particle beam. The particles of the beam (e.g., protons, electrons, and neutrons) interact with the nuclei of ISOL target constituents, producing radioisotopes through nuclear reactions, namely by fission, fragmentation, and spallation [1]. The products of the nuclear reactions first diffuse from the interior to the surface of the target and then evaporate/sublimate. ISOL targets shall operate at high temperatures in order to minimize the diffusion and releasing times of the radioisotopes. Next, the isotopes effuse to a mass separator, where they are ionized and separated based on their mass-to-charge ratio. They are finally transported to the application point. The RIB intensity, I_RIB_, depends on the particular nuclear production cross-section of the desired isotope i, σ_i_(E*), by the interaction of a particle of specific energy (E*) with the target nucleus, the number of target nuclei, N_target_, the driver beam intensity, I*, and the overall efficiency ε (depending on the diffusion, releasing, effusion, ionization, separation and transport efficiencies).
I_RIB_ = σ_i_(E*) × N_target_ × I* × ε (1)

ISOL targets can be in the form of pressed pellets, metallic foils, liquid metals, or fibers [2], with most of the target materials being refractory metals or ceramics, as oxides or carbides [3]. This method allows the production of a wide variety of radioisotopes through nuclear reactions [4]. The isotopes to be produced and the availability of the high-intensity/high-energy beam determine the choice of the nuclear targets. However, bulk, micrometric, UC_x_-based targets are the current reference at most of the ISOL facilities [5]. ISOLDE (Isotope Separator On Line Device), at CERN, is an ISOL facility where uranium carbide targets interact with a 1.4-GeV pulsed proton beam to produce radioactive nuclei for a number of different applications including: Atomic physics, nuclear physics, solid state physics, materials science, astrophysics, and medicine [6].

Uranium dicarbide, UC_2−y_, is the best candidate target material due to the high fission yields in nuclear reactions, excellent stability at elevated temperatures, and large thermal conductivity [7]. It provides large yields for neutron-deficient isotopes of heavy elements, such as Ac, Ra, Fr, At, and Rn, as well as for neutron-rich radioisotopes with 70 < A < 160, where “A” represents the mass number [8]. For a long time, uranium dicarbides have been prepared by different techniques, with each of them having advantages and disadvantages.

For example, uranium carbide target disks with 14 mm in diameter and 1 mm of thickness were prepared from the metal oxide mixed either with an excess of graphite (U:C molar ratio of 1:4) or diphathalocyanine (U:C molar ratio of 1:13) [9]. The study of their behavior under pulsed beams showed that the release of Li and Na were fast with little decay losses. The production yield of ^225^Th is about five times greater from this target (16.4 mb), when compared to identical thorium carbide targets (3.35 mb). The preparation of a AnC_x_ (An = uranium, thorium) target with a carbon backbone from molecule of diphtalocyanine has also been reported [10]. The final product was a highly porous, brittle and granular material. Its metal content was ~30–50 weight %, with the metal atoms being isotropically distributed inside the carbon matrix. The target, which has low density, 1.1–1.3 g/cm^3^, and a low content of the metal atoms (18–20 carbon atoms per metal atom), did not sinter even after 100 h at 2573 K, probably due to the high carbon concentration.

UC_x_ targets for the SPIRAL 2 and ALTO projects were described by Bajeat et al., with aim to more than 10^13^ fissions/s with a good release time, but it was difficult to keep the targets at high temperatures (>2273 K) for long periods of time [11]. Target disks around 1 mm thick, composed of graphite-embedded 20 µm–30 µm grains of uranium carbide were manufactured by compressing a mixture of uranium oxide and graphite powders. These were heated it at 2273 K under vacuum for carboreduction by the following reactions:UO_2_ + 6 C → UC_2_ + 2 C + 2 CO (2)
UO_2_ + 6 C → UC + 3 C + 2 CO (3)

It was observed that excess graphite limits the grain size of uranium carbides and minimizes the diffusion paths. Release of Cs and Rb were reported to be better from such low-density carbide targets than from high-density ones. However, Lhersonneau et al. [12] described in a review on the performance of high-density UC_x_ targets with grain sizes of ~200 µm, developed in the IRIS facility at Gatchina (Russia), that they show an excellent Cs and Rb production performance, with a yield higher than the low-density, 25 µm–35 µm targets in use at PARRNe-Orsay. Nevertheless, a slower release was observed, which was surpassed by new, high-density, UC_x_ materials with grain sizes of ~20 µm. Uranium carbide target materials with 5 µm and 20 µm grain sizes were prepared by powder metallurgy [13]. They were tested online between 2073 K and 2373 K, showing a Fr production yield very close for both materials, whereas for the Cs production the 5 µm grain target displayed a yield approximately 3 times higher than the 20-µm grain target. All these studies corroborated the convenience of using small grain size materials for ISOL targets.

A thorough study including nano-grained UC_x_ materials production for ISOL targets was undertaken within the CERN-MEDICIS project [14]. Its main objective was to increase the RIBs intensity by two or three orders of magnitude. In the light of the reasoning reported above, such an ambitious goal can be achieved not only by preparing chemically suited target materials, but also by keeping the grain sizes in the nanometric size, even at high working temperatures. However, conventional preparation methods of target materials lead to final morphologies where the grains can easily grow along the 3 directions (x, y, and z). Therefore, the development of innovative target material preparation methods, which would minimize the grain growth, or at least allow the target material to evolve only along one of the directions, is highly desirable. One possible approach of this kind can consist of the development of target materials in the form of fibers or, more specifically, micro- or nanofibers. In this case, the x and y dimensions are in the µm or nm scale, while the z dimensions can be much larger. A high porosity and a large surface-to-volume ratio in the fiber-based materials can result in a quick release of radioisotopes [15]. Fibers are obtained from polymers, their entanglement properties allowing the preparation of very long and thin samples. The diameter of the polymer fiber are in the µm–nm scale, with few contact points between them that inhibit the rapid growth and sintering at high temperatures, and, at the same time, foster in-grain isotope diffusion. Moreover, this approach permits to know the carbon and metal content in the fiber samples, offering a good control over the core density. Elemental C can be produced during the pyrolysis of a polymer [16,17], resulting in an excess of carbon in the fibers. This gives improved mechanical stability and high thermal conductivity [18,19]. The excess of carbon also inhibits grain growth and coalescence of neighboring grains [16]. Nuclear ceramic materials have been successfully prepared from polymers using the polymer infiltration and pyrolysis method [20]. There are several techniques to obtain such kind of fiber samples containing f-block elements, e.g., drawing [21], template synthesis [22], phase separation [23], self-assembly [24], and electrospinning [25]. The first four techniques are time-consuming and require the control of numerous factors (e.g., viscoelastic stress, electrical conducting properties, and dissolution). Electrospinning is a top-down technique based on the electrospinning of a polymer in solution [26] that has been used in the mass production of long, continuous, fibers [25]. In the electrospinning process, a high voltage between two electrodes is used to generate a charged jet of a polymer solution (that can contain metal(s) ion(s)) out of a needle orifice. Without any voltage the solution is held by its surface tension at the tip of a needle, which also constitutes one of the electrodes. When high voltage is applied between the needle and the collector (the other electrode), the solution is submitted to a high electric field and an induction charge appears on the surface of the liquid. A hemispherical shape is formed at the tip of the needle due to the mutual charge repulsion, which acts oppositely to the surface tension. With increasing voltage, the hemispherical surface elongates and transforms into a cone, known as the Taylor Cone [27]. By further increasing the electric field, a critical value is reached, where the repulsive force overcomes the surface tension and the fluid jet is ejected from the tip of the Taylor cone. After the polymer solution discharging, a bending instability followed elongation process occurs. On the way to the collector, the solvent evaporates and solid fibers are recovered.

This work is focused on the electrospinning preparation of uranium carbide fibers, with micrometer diameter but composed of nanometer size grains. Polymeric fibers with uranium salts were prepared by the electrospinning technique, and subsequently heat-treated in two steps. A first stage, until 823 K, was aimed at decomposing the base polymer, and a second step, at 2073 K promoted the carboreduction reaction. The materials obtained after each of these steps were characterized in terms of crystal structure, morphology, grain size, and composition.

## 2. Materials and Methods 

The experimental section is divided into three sub-sections. In the first, the preparation of the viscous solutions with uranium precursor salts dissolved is briefly explained. In the second, the electrospinning production of fibers from the viscous solutions is described; in addition, heat treatments of the as spun fibers at medium temperatures (to decompose the polymer) and at high temperatures (to promote the carboreduction of the UO_2_) are discussed thoroughly. Finally, the last sub-section depicts the characterization techniques employed to study the fiber samples.

### 2.1. Preparation of the Viscous Solution with the Uranyl Precursors

Uranium (primary isotope ^238^U) is a weak α-emitter (4.197 MeV) with a half-life of 4.47 × 10^9^ years. Manipulations and reactions should be carried out in monitored fume hoods in a radiation laboratory equipped with α- and β-counting equipment. Three different uranium precursor salts have been used, namely uranyl acetylacetonate, uranyl acetate, and uranyl formate. Uranyl acetylacetonate can be prepared by the slow addition of aqueous sodium hydroxide to solutions containing uranyl nitrate and acetylacetone, while uranyl formate is obtained from uranyl nitrate solutions with an excess of formic acid [28,29]. Uranyl acetate is commercially available (Merck, Darmstadt, Germany), but can also be prepared by reacting UO_3_ with acetic acid [30]. The cellulose acetate polymer (Sigma-Aldrich, St. Louis, Missouri, USA, no purity info, average Mn ~50,000 by GPC) was first dissolved in a a:b (*v/v*) acetic acid (PanReac, Barcelona, Spain, Acetic Acid glacial, 99.5%) and 2,4-pentanedione solution (Fluka Chemie AG, Buchs, Switzerland, acetylacetone p.a.) at 333 K under magnetic stirring conditions (a:b = 1:2 for the case of uranyl acetate and 1:1 for uranyl acetylacetonate and uranyl formate precursors). After the cellulose acetate dissolution, the uranium precursor salt was added to the solution in three different U/C mole ratios, 1/2, 1/4, and 1/6, taking into account the polymer matrix and uranyl anions. The mixture was stirred until all uranyl salt was completely dissolved, with the final volume of the solution being adjusted to a polymer concentration of 10, 12.5, or 15 weight % by adding or evaporating the solvent (which was controlled by weighting). After that, the viscous solutions were ready to be electrospun.

### 2.2. Preparation and Heat-Treatment of the Fibers

There are three components in an electrospinning system namely (i) a high voltage power supply, (ii) a pump with a syringe containing a metallic needle with small diameter, and (iii) a metal collector. The syringe filled with the viscous solution was placed on the pump and the liquid was ejected with a rate between 0.8–2.5 mL/h. A high voltage of 18–30 kV was used to generate a charged jet of polymer solution out of the needle orifice, which was recovered in an aluminum-foil collector placed at ~12 cm away from the needle tip. The as-spun fibers were then heated to 823 K at 1 K/min heating rate inside a tubular furnace under an argon gas flow atmosphere. After that, the decomposed material was inserted into a carbon glass crucible, which was placed inside an induction furnace chamber that was vacuum pumped. Once the 2 × 10^−6^ mbar pressure has been reached, the heat treatment started. The temperature was increased under vacuum (not letting surpass 2 × 10^−5^ mbar) to promote the carbo-thermal reaction and the extraction of the CO produced. At the end, the decomposed material was heated at 2073 K for 2 h.

### 2.3. Characterization

The morphology and chemical compositions of the as-spun fibers and heat-treated materials were studied using optical, scanning and transmission electron microscopy (SEM and TEM respectively). SEM observations were made using a JEOL 7001F microscope (JEOL Ltd., Tokyo, Japan) operating at 20 kV, with Oxford light elements EDS detector (Oxford Instruments, Concord, MA, USA), and a Hitachi S2400 system (Hitachi, Tokyo, Japan) working at 15 kV, with Bruker light elements EDS detector (Bruker Corporation, Billerica, MA, USA). A TecnaiG2 TEM 200 kV with an EDS analysis system and a high-angle annular dark-field (HAADF) detector for scanning transmission electron microscope (STEM) imaging (Thermo Fisher Scientific, Eindhoven, The Netherlands), equipped with a Gatan™ Tridiem GIF camera (Gatan, Pleasanton, CA, USA), was used to characterize the crystallite sizes and morphology. The samples were prepared by crushing tiny fragments of the various compounds in methanol. The suspension was allowed to decant, and a droplet was subsequently deposited on a copper grid coated with carbon. 

Heat-treated samples were manually crushed in a mortar grinder, the resulting powder was placed onto a low-noise Si single crystal sample holder and studied by powder X-ray diffraction (XRD). XRD measurements were performed in a PANalytical X’Pert-Pro diffractometer (PANalytical Co., Almelo, Holland, Cu K_α_-radiation) with Bragg-Brentano geometry. Step-scanning mode XRD patterns were taken in the 10–100° 2θ region using the θ/2θ configuration, a step size of 0.04° and a counting time of 40 s per step. The theoretical powder patterns were simulated with the help of the Powder-Cell software [31] and the lattice parameters were calculated by using the UnitCell software [32]. 

Thermogravimetry analysis (TGA) of the electrospun material was performed from room temperature up to 873 K using a Dupont 951 Thermo-Gravimetric Analyser (TA Instruments, New Castle, PA, USA) equipped with a Temperature Programmer Interface TA Controller System. The measurements were carried out under nitrogen, with a constant flux of 50 mL/min and a heating rate of 10 K/min. 

Raman spectra were collected with a Jobin-Yvon^®^ T64000 spectrometer (HORIBA JobinYvon, Longjumeau, France) used in the single spectrograph configuration. The 647 nm line of a Kr^+^ Coherent^®^ laser was used as an excitation source, with a nominal power at the laser cavity of 100 mW. This wavelength and power were chosen in order to optimize the signal/noise ratio and reduce undesirable oxidation/burning effects on the sample surface [33]. The Raman spectrograph was calibrated with the T_2g_ excitation line of a silicon single crystal, set at 520.5 cm^−1^ [34]. Raman spectra were recorded with an instrumental uncertainty of ±1 cm^−1^.

## 3. Results and Discussions

The feasibility of the precursor fibers formation as a function of the cellulose acetate polymer concentration, determined by optical microscopy observation, is shown in Table 1 for the three different uranyl salts. The quality of as-spun uranium-containing fibers strongly depends on the amount of the polymer present in the viscous solution, as the polymer entanglement property can keep the fibers continuous during their migration from the needle tip to the collector. Concurrently, the solvent evaporates and dry fibers can be collected on the aluminum-foil collector.

SEM observations confirmed the optical microscopy results. For the 10% weight concentration of cellulose acetate polymer, the spun material consists on solidified drops for any of the applied voltages and uranyl salts. The formation of continuous fibers is inhibited for low polymer concentrations due to the lack of viscosity of the solution, resulting in the formation of droplets, which can be connected by very thin fibers (average diameter of ~0.1 µm), along with beads (Figure 1a,b). The solidified droplets are concave in shape and can have one or more holes. Increasing the cellulose acetate concentration, and for voltages between 18 and 22 kV, a web consisting of beads interconnected by ~1 µm diameter fibers is formed, as can be seen for the 12.5% weight concentration (Figure 1c,d). At higher concentrations of cellulose acetate polymer, i.e., 15%, the fibers obtained from electrospinning are of good quality in the 18–30 kV voltage range for all uranyl salts, but the uranyl acetate gives fibers of better quality. The fibers obtained are thin, with diameters ranging from 0.5 to 2 µm and a smooth surface, without beads and droplets, as shown in Figure 1e–g. Above the 15% concentration, the solution becomes too viscous to be electrospun.

TGA results for cellulose acetate and the as-spun fibers of uranyl salts (acetate (ac), acetylacetonate (acac) and formate) are shown in Figure 2. Three distinct weight loss stages can be identified in all precursor fibers. At the beginning, from room temperature to ~333 K, the weight loss is ~6% for all samples, which is mainly due to solvents evaporation. Next, a flat plateau is seen, until ~503 K for the uranium materials and ~613 K for the cellulose acetate polymer. Between ~503 K and ~573 K a second weight loss drop, corresponding to more than 50% decrease, is observed for uranyl acetylacetonate and uranyl formate precursor fibers. In the case of uranyl acetate, this drop also exists, but at higher temperatures, between ~523 K and ~633 K, while for cellulose acetate it goes from ~613 K to ~663 K. This large weight decrease is due to the decomposition reaction, where the uranyl salts and cellulose acetate are converted mainly into uranium oxide, UO_2_, and graphite, C (see SEM/EDS and XRD results below). The higher decomposition temperature of the uranyl acetate precursor is probably due to the presence of strong interactions between the acetate groups of cellulose acetate and uranyl salt. Thereafter, no further decomposition was noticed for the cellulose acetate polymer, but a smooth weigh decrease, which corresponds to the decomposition of the remaining organics, with the formation and subsequent evaporation of volatile compounds, was observed for the uranium materials. In this stage, the loss is more pronounced for the uranyl formate fibers, which can be attributed to the volatile nature of the formate ligands. This last step corresponds to a weight loss of ~10% and finishes at ~793 K.

Following the electrospinning procedure and with the data obtained from TGA, the as-spun fibers were decomposed in a tubular furnace. The fibers were inserted in the furnace that was first settled at 296 K, and then heated with a rate of 1 K/min up to 823 K in a steady argon-gas flow. The release of smoke-type gases was observed, the decomposition being considered complete when the gas release stopped. With this procedure, it was possible to retain the fiber morphology after the heat treatment (Figure 3a–c). In contrast, if the fibers were heated rapidly, the polymer melted before the decomposition and the original shape was lost. A significant decrease of the fibers diameter is observed, which accounts for the large weight loss indicated by TGA. EDS analysis indicated that the decomposed material is constituted by uranium, carbon, and oxygen.

The decomposed material was further heated under vacuum at 2073 K for 2 h, to promote the uranium oxide carboreduction. The final material is composed of highly porous and uncompact fibers, as shown in Figure 4. 

The pores may form due to inward carboreduction reactions that release carbon oxide gases. EDS analysis indicate that the carbo-reduced material contains uranium, oxygen, and carbon in excess (>60 in atomic %), which points to an uncompleted carboreduction reaction.

X-ray patterns of the decomposed and heated at 2073 K samples are presented in Figure 5a,b. The fibers decomposed at 823 K are mainly constituted by uranium dioxide, the a = 5.430(1) Å small lattice parameter pointing to a hyperstoichiometric oxygen composition [35]. Carbon is also present, but its small electron density and the possible existence of disorder only allowed the detection of its most intense peak. The UO_2+x_ peaks are very broad, indicating a nanocrystalline character of the decomposed fibers. The material heat treated at 2073 K for 2 h has the tetragonal UC_2−y_ as the major phase, but UO_2+x_ still exists in a moderate concentration (~30 vol. %), which points to the uncompleted carboreduction reaction and agrees with the EDS results.

An increase of the UO_2+x_ lattice parameter is observed (a = 5.457(1) Å) together with an increase of the sharpness of the uranium oxide peaks, pointing to a decrease of the hyperstoichiometry and increase of crystallinity. The UC_2−y_ lattice parameters (a = 3.5144(3) Å, a = 5.970(1) Å) are in agreement with published data [36]. The incomplete carboreduction reaction can be partially explained by the excess of carbon that impedes the atoms diffusion. However, it is expected that longer heat treatments would allow the total oxide carboreduction into uranium dicarbide.

Transmission electron microscopy (TEM) images of the decomposed fiber and carbo-reduced material are shown in Figure 6a,b. The decomposed material is nanocrystalline, with average grain sizes below 10 nm, being composed of uranium oxide and carbon, which supports the X-ray diffraction results. TEM observations also reveal that in the decomposed fibers the uranium oxide nano-particles are embedded in a graphite matrix. After the heat-treatment, most of the uranium oxide was reduced to uranium dicarbide, which has particle sizes in the range 4 nm–10 nm.

Raman spectroscopy studies have been carried out on the final carbo-reduced material. A representative spectrum is shown in Figure 7.

The spectrum displays no clear features at energies below 1000 cm^−1^. Yet, one can spot Raman bands related to the presence of moderate amounts of UO_2+x_ (at 445 cm^−1^, 575 cm^−1^, and 630 cm^−1^ [35]). However, they are rather blurred and probably close to the current intensity detection limits. Two large peaks are observed in the spectrum at higher wave numbers, the first being located at ~1320 cm^−1^ and the second at ~1590 cm^−1^. As no Raman active peaks exist for uranium dicarbide, the observed ones can only be due to carbon. Disordered and ordered graphite display D and G bands, located at ~1350 cm^−1^ and ~1580 cm^−1^, respectively [37]. The G band is originated from the E_2g_ vibration of ordered graphite, corresponding to the relative motion between two sp^2^ carbon atoms. The D band is due to the A_1g_ breathing modes of sp^2^ ring atoms. This Raman-active vibration is symmetrically forbidden for a perfect graphite crystal, but becomes quite intense in a defective lattice. Comparison with the present results indicates that the first peak corresponds to the D band, while the second is certainly linked to the G band. The energy shifts observed in the peak positions with respect to the nominal values expected for pure graphite are attributable to the specific morphology and chemical environment of the present materials. The higher intensity of the D band confirms the existence of considerable disorder in the graphite, as was already suggested by the X-ray diffraction results.

## 4. Conclusions

Uranium carbide fibers with micrometer diameters and nanometer grain sizes were successfully synthesized from viscous solutions of cellulose acetate and uranyl salts using (i) the electrospinning method, followed by (ii) subsequent decomposition by slow heating up to 823 K under argon and (iii) carboreduction at 2073 K under vacuum for 2 h.

Good quality precursor fibers, with diameters ranging from 0.5 to 2 µm and smooth surfaces, were obtained for all uranyl salts solutions having 15% weight concentrations of cellulose acetate, indicating that a critical factor for the fibers formation is the polymer concentration, although the best ones were achieved from the solutions containing uranyl acetate. A slow heating rate, of 1 K/min, during the decomposition step allowed to retain the fiber morphology up to high temperature, the samples being composed by UO_2_ and C with average grain sizes below 10 nm. The release of volatile compounds and crystallization during the decomposition process lead to a decrease of the fibers diameter and the formation of such nanostructured material. The high-temperature heat treatment at 2073 K promotes the carboreduction of UO_2_ to UC_2−y_ while keeping the grain sizes in the 4 nm–10 nm range, but the reaction was not completed after 2 h, with a moderate volume concentration of oxides still existing. A carbon excess and the fibers morphology, which hinders the atoms diffusion due to the small number of contact points between the carbon and uranium oxide grains, are probably the reason for such incomplete carboreduction. However, longer heat treatments are expected to induce a total carboreduction of the oxide into uranium dicarbide.

The first preliminary online tests of the uranium carbide fiber materials are expected to be performed at ISOLDE, after the CERN Long Shutdown 2, scheduled for the middle of 2021.

## Figures and Tables

**Figure 1 nanomaterials-10-02458-f001:**
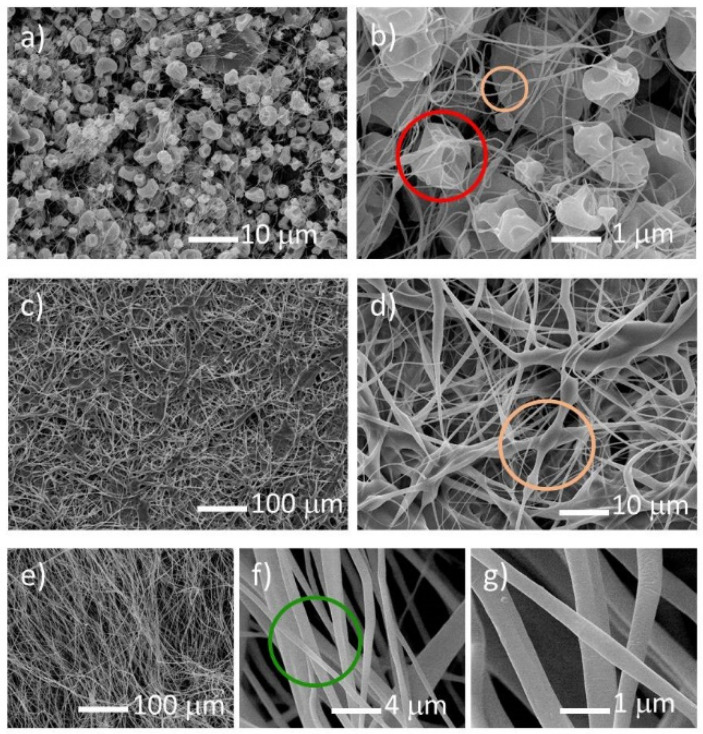
Scanning electron microscopy images of uranyl acetate as-spun fibers with U/C = 1/6 and a cellulose acetate polymer weight concentration of 10% (**a**,**b**), 12.5% (**c**,**d**), and 15% (**e**–**g**).

**Figure 2 nanomaterials-10-02458-f002:**
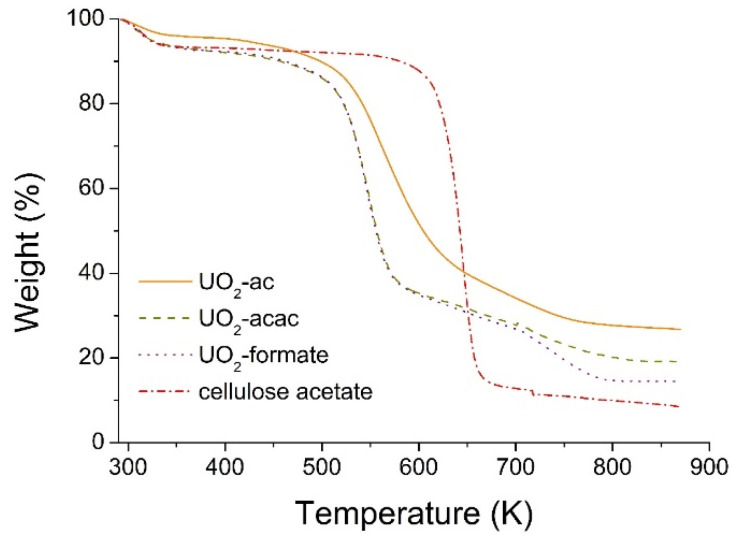
Thermal gravimetric curves of cellulose acetate, and the uranyl salts as-spun fibers with U/C = 1/4 and a cellulose acetate weight concentration of 15%.

**Figure 3 nanomaterials-10-02458-f003:**
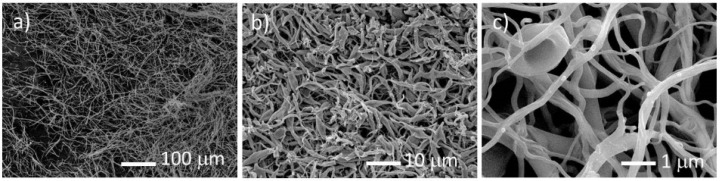
Scanning electron microscopy images of the decomposed fibers of (**a**) uranyl acetate, (**b**) uranyl formate, and (**c**) uranyl acetylacetonate salts with 15% cellulose acetate weight concentration and U/C = 1/6 after heat treated at 823 K.

**Figure 4 nanomaterials-10-02458-f004:**
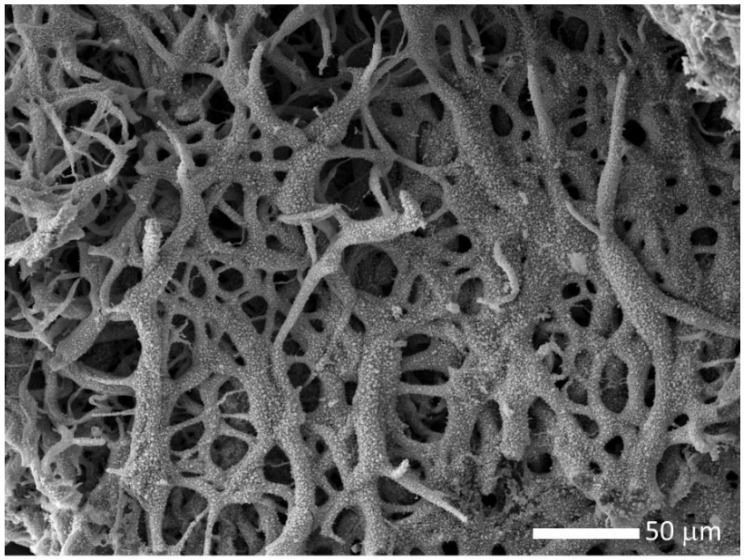
Scanning electron microscopy image of the fibers heat treated at 2073 K for 2 h of the decomposed fibers of uranyl acetate salt with cellulose acetate polymer concentration (*w/v*) 15% of U/C = 1/4.

**Figure 5 nanomaterials-10-02458-f005:**
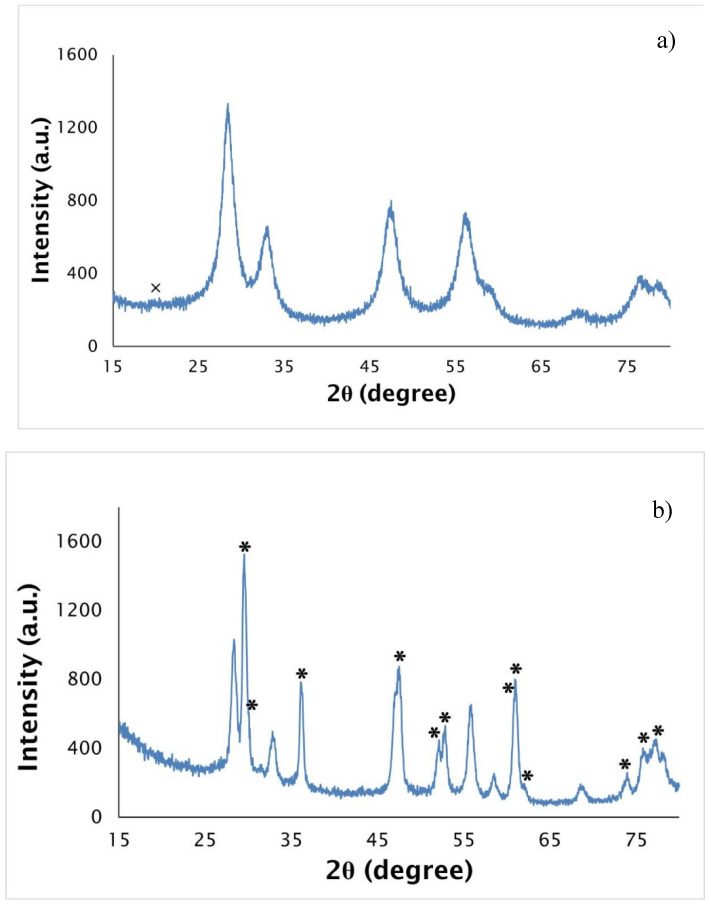
X-ray diffractograms of the uranyl acetate as-spun fibers with U/C = 1/4 and a cellulose acetate polymer weight concentration of 15% after (**a**) decomposed at 823 K and (**b**) after heat treated at 2073 K for 2 h (crosses—carbon, stars—UC_2_, unmarked peaks—UO_2_).

**Figure 6 nanomaterials-10-02458-f006:**
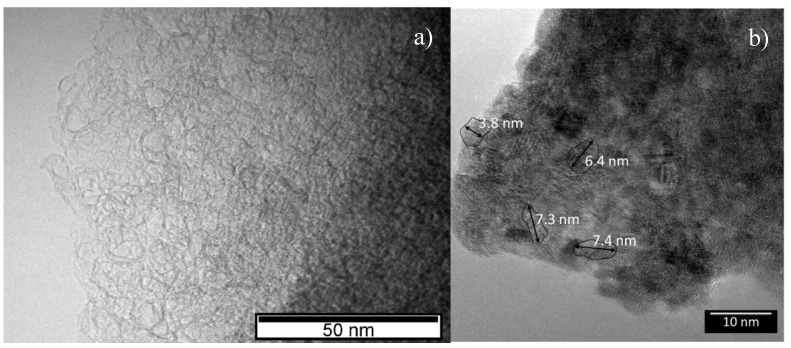
Transmission electron microscopy images of the uranyl acetate as-spun fibers with U/C = 1/4 and a cellulose acetate polymer weight concentration of 15% after (**a**) decomposed at 823 K and (**b**) after heat treated at 2073 K for 2 h.

**Figure 7 nanomaterials-10-02458-f007:**
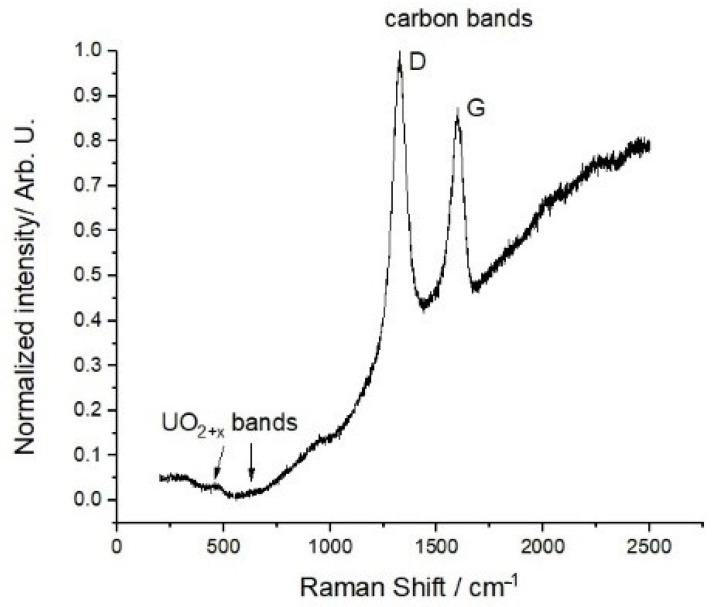
Raman spectrum of the uranyl acetate as-spun fibers with U/C = 1/4 and a cellulose acetate polymer weight concentration of 15% after heat treated at 2073 K for 2 h.

**Table 1 nanomaterials-10-02458-t001:** Feasibility of precursor fibers formation for the three different concentrations of cellulose acetate polymer, along with the various uranyl salts.

Salt Polymer Weight (%)	Uranyl Acetate	Uranyl Acetylacetonate	Uranyl Formate
10	No	No	No
12.5	Yes	Yes	Yes
15	Yes	Yes	Yes

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
