# Peer review of "Uranium Carbide Fibers with Nano-Grains as Starting Materials for ISOL Targets"

_nanomaterials, 2020, doi:10.3390/nano10122458_

Round 1

Reviewer 1 Report

The manuscript is very well written.  The authors provide the motivation for why they are developing the materials, and what the desirable properties for the materials would be.  They give a detailed description of the production process as well as the materials characterization methods used.  The paper was easy to read, as the authors justified the different materials and processes at each step.  It will be interesting to see how the new materials work as a target for the radioactive beams experiments that are scheduled for next year.  Well done.

Author Response

We thank the Reviewer for the kind words.

Reviewer 2 Report

Dear Authors,

in your interesting manuscript, the following points should be added/changed to further improve it:

  • Author names: Affiliation 4 must be added after T. Stora.
  • Abstract: You mention three concentrations of which the highest is the best. What about even higher concentration - which may be even better -, were they also tested?
  • Eq. 1: Please use the right multiplication signs. Besides, the I* looks different here as in line 41, as if the * were superscripted.
  • line 169: Which heating rate is meant by "slowly"?
  • Fig. 5 is prepared with Excel, while the other figures are made in Origin. Is it possible to use Origin also for Fig. 5, just due to optical reasons?
  • line 301: "which"
  • Generally, since most people will probably use electrospinning for preparation of less "special" materials, it would be great if you could add a few sentences about the necessary safety regulations when working with uranium salts.

Author Response

We thank the Reviewer for the comments, which are answered point-by-point below.

  • Author names: Affiliation 4 must be added after T. Stora.

Affiliation 4 was added after T. Stora.

  • Abstract: You mention three concentrations of which the highest is the best. What about even higher concentration - which may be even better -, were they also tested?

Higher cellulose acetate polymer concentrations were tested, but their viscosity was too high to be electrospun. This is mentioned in page 6, line 232.

  • Eq. 1: Please use the right multiplication signs. Besides, the I* looks different here as in line 41, as if the * were superscripted.

The right multiplication signs are now in Eq. 1. The I* in line 41 and in Eq. 1 are now equal.

  • line 169: Which heating rate is meant by "slowly"?

Slowly means that the temperature was increased to promote the carbo-thermal reaction and the extraction of the CO produced, but not letting surpass 2 × 10–5 mbar. The sentence was rephrased to clarify this point (lines 175-176)

  • Fig. 5 is prepared with Excel, while the other figures are made in Origin. Is it possible to use Origin also for Fig. 5, just due to optical reasons?

We agree to do it, if the publisher advises the change.

  • line 301: "which"

Corrected

  • Generally, since most people will probably use electrospinning for preparation of less "special" materials, it would be great if you could add a few sentences about the necessary safety regulations when working with uranium salts.

A few sentences about the necessary safety regulations when working with uranium salts were included in the experimental section.

Reviewer 3 Report

The paper "Uranium carbide fibers with nano-grains as starting materials for ISOL targets" by Sanjib Chowdhurry has been submitted for publication in Nanomaterials.

The paper describes the preparation of uranyl carbide containing fibers by electrospinning of polymer solution followed by thermal treatments. The uranyl carbide fibers with small grain size will be used as ISOL targets at the CERN Isolde facility by mid 2021.

The paper is well planned, experimental results are clearly described and exhaustively discussed. Conclusions are soundly.

It is my opinion that the paper can be published after minor revisions, listed in the following:

in the review of exhisting literature, the authors should mention and discuss the 4 papers published by Singh, A.K. et al. in 2006-2008 about the preparation of polymer nanofibers containing uranium carbide;

line 25: "with particle size ...". The authors refer to the grain size uf UC2-y phase. From on-line Cambridge dictonary, particle is "a very small piece of something", which gives the idea on individual particles. Please rephrase the sentence;

line 74: "difficulty", probably the adjective "difficult" would be more appropriate;

line 108: "The grains of the polymer fibre ...". The term does not properly apply to polymer morphology, where you speaks about crystallites. Probably the authors would mean diameter of the fibres!

line 119: "based on the electrospun of ...". The appropriate word would be electrospinning, even tough it is an unpleasant repetition. Please rephrase the sentence;

line 130: "bending instability followed ...". Please verify the sentence for inproved readability.

line 153: "under magnetic stirrer condition", please change in "under magnetic stirring condition" or "using magnetic stirrer";

line 156: "different U/C ratios". Please specify if the weight or the mole ratio is considered, and if C is calculated taking into account the polymer marix and uranyl anions, thereby discarding the solvent;

line 157: how is the polymer concentration measured or taken into account during solvent evaporation?

line 169: the formation of CO and CO2 is referred too. Please include in the reaction scheme reported at lines 80-81 the reaction(s) responsible for CO2 formation;

figure 1: please verify magnification reported on the pictures. In particular, pictures g & h are illustrtated at the same magnifications? Also, confirm the magnification for c and f images;

line 219-220: "The droplets ... than the beads". Please rephrase the sentence: droplets have to be described as particles or solified droplets. The overall particle shape cannot be concave, and abviously the number of holes is higher than the number of particles, if there are 1 or more holes per particle;

line 234: the evolution of adsorbed gas is not detectable with TGA technique;

line 246: "evaporation of volatile compounds". The term volatile is pleonastic;

line 248: "in a weight loss of 10%". The sentence is quite misleading, since the residue is 10% at this point. Actually the name of ordinate axes in fig 2 is misleading. Better to change in Weight Residue (%);

line 269: "The pores may form due to inward carboreduction ...". No pores are visible of the fibres surface. The pores and voids exhists between the fibers, therefore are due to electrospinning process and not to decomposition reactions;

line 288-290: The sentence should be rephased or eliminated. In fact, the grain size is at least an order of magnitude smaller than fiber diameter. Therefore, the grains are completely embedded. Indeed, the authors should specify in the experimental section if the uranyl salts completely dissolve in the polymer solution, resulting in an homogeneous solution;

line 290: "difficults" should be changed in "hinders" or "impedes";

figure 6: can the image quality and contrast be improved?

line 301: "whuch" should be changed in "which";

line 312: in the identification of Raman peaks it should be considered that at the end of carboreduction reaction there still exhist a percentage of residual UO2. Therefore, peaks possibly related to this compound should be considered in the discussion of Raman spectra.

ded as

Author Response

We thank the Reviewer for the detailed analysis and comments, which are answered point-by-point below.

in the review of existing literature, the authors should mention and discuss the 4 papers published by Singh, A.K. et al. in 2006-2008 about the preparation of polymer nanofibers containing uranium carbide;

A report, "Novel Processing of Unique Ceramic-Based Nuclear Materials and Fuels", summarizing all those papers and also proccedings and conference presentations, covering the period between April 1, 2005 and August 31, 2008, was compiled by Raman P. Singh. This report describes the polymer infiltration and pyrolysis method that was used for the fabrication of ceramic based materials and is now mentioned in the "Introduction".

line 25: "with particle size ...". The authors refer to the grain size uf UC2-y phase. From on-line Cambridge dictonary, particle is "a very small piece of something", which gives the idea on individual particles. Please rephrase the sentence;

The sentence was rephrased in agreement with the Reviewer suggestion.

line 74: "difficulty", probably the adjective "difficult" would be more appropriate;

We agree with the Reviewer and have corrected accordingly

line 108: "The grains of the polymer fibre ...". The term does not properly apply to polymer morphology, where you speaks about crystallites. Probably the authors would mean diameter of the fibres!

Corrected

line 119: "based on the electrospun of ...". The appropriate word would be electrospinning, even tough it is an unpleasant repetition. Please rephrase the sentence;

Done

line 130: "bending instability followed ...". Please verify the sentence for improved readability.

Done

line 153: "under magnetic stirrer condition", please change in "under magnetic stirring condition" or "using magnetic stirrer";

Done

line 156: "different U/C ratios". Please specify if the weight or the mole ratio is considered, and if C is calculated taking into account the polymer marix and uranyl anions, thereby discarding the solvent;

Done

line 157: how is the polymer concentration measured or taken into account during solvent evaporation?

It was taken into account by weighting. This is now refered in the manuscript.

line 169: the formation of CO and CO2 is referred too. Please include in the reaction scheme reported at lines 80-81 the reaction(s) responsible for CO2 formation;

We thank the Reviewer for this comment, as CO2 is not formed by the carboreduction reaction of UO2. This mistake was corrected.

figure 1: please verify magnification reported on the pictures. In particular, pictures g & h are illustrtated at the same magnifications? Also, confirm the magnification for c and f images;

We thank the Reviewer for this comment and indeed some scales were wrong. This is now corrected.

line 219-220: "The droplets ... than the beads". Please rephrase the sentence: droplets have to be described as particles or solified droplets. The overall particle shape cannot be concave, and abviously the number of holes is higher than the number of particles, if there are 1 or more holes per particle;

We thank the Reviewer for this comment. The sentence was corrected.

line 234: the evolution of adsorbed gas is not detectable with TGA technique;

We thank the Reviewer for this comment. The sentence was corrected accordingly.

line 246: "evaporation of volatile compounds". The term volatile is pleonastic;

Albeit agreeing that the term "volatile" can be pleonastic, we think that we should keep it to stress the evaporation process of such compounds.

line 248: "in a weight loss of 10%". The sentence is quite misleading, since the residue is 10% at this point. Actually the name of ordinate axes in fig 2 is misleading. Better to change in Weight Residue (%);

The sentence was clarified. The name of ordinate axes in fig 2 was changed to "Weight".

line 269: "The pores may form due to inward carboreduction ...". No pores are visible of the fibres surface. The pores and voids exhists between the fibers, therefore are due to electrospinning process and not to decomposition reactions;

Very small pores exist in the fibers surface (see Figure 4) due to the release of CO during the carboreduction reaction.

line 288-290: The sentence should be rephased or eliminated. In fact, the grain size is at least an order of magnitude smaller than fiber diameter. Therefore, the grains are completely embedded. Indeed, the authors should specify in the experimental section if the uranyl salts completely dissolve in the polymer solution, resulting in an homogeneous solution;

The sentence was rephrased accordingly. It was specified in the Experimental section that the uranyl salts were completely dissolved in the solution.

line 290: "difficults" should be changed in "hinders" or "impedes";

Done

figure 6: can the image quality and contrast be improved?

We have tried, but no improvement was obtained. Therefore, the contrast was left as it was.

line 301: "whuch" should be changed in "which";

Done

line 312: in the identification of Raman peaks it should be considered that at the end of carboreduction reaction there still exhist a percentage of residual UO2. Therefore, peaks possibly related to this compound should be considered in the discussion of Raman spectra.

We thank the reviewer for this remark. We addressed peaks possibly related to the presence of residual UO2 in lines 311-313.